# Fetal and Neonatal Outcomes in Fetuses with an Estimated Fetal Weight Percentile of 10–20 in the Early Third Trimester: A Retrospective Cohort Study

**DOI:** 10.3390/diagnostics15172251

**Published:** 2025-09-05

**Authors:** Miguel A. Mendez-Piña, Mario I. Lumbreras-Marquez, Sandra Acevedo-Gallegos, Berenice Velazquez-Torres, Maria J. Rodriguez-Sibaja, Dulce M. Camarena-Cabrera, Juan M. Gallardo-Gaona

**Affiliations:** 1Maternal-Fetal Medicine Department, Instituto Nacional de Perinatologia, Mexico City 11000, Mexico; miguel.mepm@hotmail.com (M.A.M.-P.); mario.lumbreras@inper.gob.mx (M.I.L.-M.); dracevedo_sandra@yahoo.com.mx (S.A.-G.); berevelazquez70@yahoo.com.mx (B.V.-T.); mariajose.rodriguezs@yahoo.com (M.J.R.-S.); dulcemariacamarena@hotmail.com (D.M.C.-C.); 2Department of Epidemiology and Public Health, Universidad Panamericana School of Medicine, Mexico City 03920, Mexico

**Keywords:** estimated fetal weight, fetal growth, percentile, neonatal outcomes, perinatal outcomes, fetal growth restriction, small for gestational age

## Abstract

**Background:** Fetal size is often dichotomized as normal or abnormal using the 10th percentile of estimated fetal weight (EFW) or abdominal circumference as a cutoff. While the risk of adverse perinatal outcomes decreases with increasing fetal weight percentile, no percentile completely eliminates that risk. **Objective:** The aim of this study was to compare perinatal outcomes between fetuses with an EFW between the 10th and 20th percentiles and those with an EFW between the 20th and 90th percentiles (i.e., >20 and <90) at the beginning of the accelerated growth stage (28.0–30.0 weeks’ gestation). **Methods:** We conducted a retrospective cohort study of singleton pregnancies managed at a quaternary center in Mexico City (2017–2024). Outcomes were compared based on EFW percentiles at 28.0–30.0 weeks. The primary outcome was adverse neonatal outcome (ANeO), defined as the presence of at least one of the following: umbilical artery pH ≤ 7.1, 5 min Apgar ≤ 7, NICU admission, early neonatal hypoglycemia, non-reassuring fetal status, respiratory distress syndrome, intraventricular hemorrhage, hypoxic–ischemic encephalopathy, or perinatal death. Secondary outcomes included progression to fetal growth restriction (FGR) and low birth weight. Modified Poisson regression was used to estimate adjusted risk ratios (aRRs) with 95% confidence intervals (CIs). **Results:** Among 650 cases, ANeO occurred in 45.8% of fetuses in the 10th–20th percentile group vs. 29.4% in the 20th–90th percentile group (aRR: 1.51, 95% CI: 1.22–1.86; *p* < 0.001). FGR and low birth weight were also more frequent in the 10th–20th percentile group (21.1% and 27.6% vs. 6.4% and 5.8%, respectively; *p* < 0.001). **Conclusions:** Fetuses between the 10th and 20th percentiles at 28–30 weeks have increased risks of neonatal morbidity, FGR, and low birth weight.

## 1. Introduction

Diagnosing growth disorders has allowed the identification of fetuses at risk of presenting adverse perinatal events, streamlining their follow-up, and potentially improving fetal and neonatal outcomes [1,2,3]. In routine obstetrics, fetal size is dichotomized as normal at or above the 10th percentile and abnormal below the 10th percentile of estimated fetal weight (EFW) or abdominal circumference (AC) [1,2]. However, although adverse perinatal outcomes are more prevalent in fetuses with abnormal fetal growth (i.e., EFW < 10th centile [4,5,6,7,8]), they are also common in fetuses with an appropriate-for-gestational-age EFW, probably because many have a certain degree of placental dysfunction even at higher EFW percentiles [8,9,10,11].

Although the concept of abnormal fetal growth is central to modern fetal medicine, there is still limited information on the EFW percentile threshold at which morbidity and mortality are significantly decreased. Several studies have proposed different reference values for this threshold [12,13,14]. Previous reports have emphasized that the risk of adverse perinatal outcomes, including stillbirth, decreases progressively as the EFW percentile increases. However, no clear percentile threshold has been identified beyond which this risk can be ruled out, making the surveillance of fetuses above the 10th percentile particularly challenging [7,11,15,16]. The recent literature suggests that the relationship between fetal size and perinatal risk follows a continuum rather than adhering to fixed cutoffs. In this context, fetuses with an EFW between the 10th and 20th percentiles—though often considered within the “normal” range—may still carry a higher risk of complications, despite not meeting diagnostic criteria for fetal growth restriction (FGR) [9]. This lack of consensus on risk thresholds limits the comparability of studies and the implementation of standardized monitoring strategies. Thus, the primary aim of this study was to evaluate adverse perinatal outcomes in fetuses with an EFW between the 10th and 20th percentiles during the early third trimester (i.e., 28.0–30.0 weeks of gestation), compared with those in the >20th to <90th percentile range, which is commonly used as a reference for normal growth. Secondary aims included comparing progression to FGR and low birth weight between these groups. We hypothesized that fetuses in the 10th–20th percentile subgroup would have a significantly higher risk of adverse outcomes, supporting the need for further investigation into this potentially vulnerable population.

## 2. Materials and Methods

This retrospective cohort study was conducted at the Maternal–Fetal Medicine Department of the Instituto Nacional de Perinatologia in Mexico City from 1 March 2017 to 31 January 2024. The study employed a frequency-matched retrospective cohort design. Fetuses with an EFW between the 10th and 20th percentile were compared to a control group with an EFW in the >20th–<90th percentile range. While participants were not individually matched, both groups were selected from the same institution during the same study period and restricted to the 28.0–30.0 weeks gestational age window at inclusion to minimize temporal- or protocol-related bias. Baseline characteristics, including maternal comorbidities, were accounted for in the adjusted statistical analyses described below. Singleton pregnancies between 28.0 and 30.0 weeks of gestation (where a fetal growth pattern associated with cell hypertrophy and accumulation of fat, muscle, and connective tissue is observed [17]) with an EFW in the 10th–20th percentile of Hadlock [18] were included for analysis as mentioned above. A corresponding control group at the same fetal growth spurt stage, consisting of fetuses with an EFW between the >20th and <90th centile (Hadlock) [18] was also included in the study (Figure 1 and Figure 2). All patients had two or more growth evaluations during pregnancy; for both groups, gestational age was confirmed by the last menstrual period, confirmed by a first-trimester ultrasound (i.e., craniocaudal length) or an early second-trimester biparietal diameter [19,20]. Pregnancies were excluded if they had incomplete clinical records; if fetuses presented with structural anomalies, chromosomal abnormalities, or congenital infections; or if FGR was diagnosed at baseline based on the international Delphi consensus criteria [21]. Likewise, cases were excluded if the fetus was classified as large for gestational age (LGA) or defined as having an EFW or AC ≥ 90th percentile at baseline. Pregnancies with fewer than two growth evaluations during follow-up were also excluded. For this study, “baseline” was defined as the first third-trimester growth ultrasound performed between 28.0 and 30.0 weeks of gestation.

All scans were performed by maternal–fetal medicine specialists using a Voluson E8, 730 Expert (GE Healthcare, Zipf, Austria) or Samsung V7 (Samsung Medison, Seoul, Republic of Korea) according to standardized guidelines for fetal biometry [1,2,22]. Fetal biometry was obtained using standard ultrasonographic measurements, including biparietal diameter (BPD), head circumference (HC), AC, and femur length (FL). EFW was calculated using the Hadlock formula, which incorporates all four parameters (HC, BPD, AC, and FL) and has been validated as one of the most accurate models for estimating fetal weight [23]. The decision to rely solely on the Hadlock formula was supported by evidence demonstrating its superior predictive accuracy in estimating birth weight compared to other published models [24]. This formula was also used to assign fetal weight percentiles based on gestational age [18]. Although various fetal growth standards exist, we used Hadlock-based percentiles due to their widespread clinical use and because this chart has been validated in our population, demonstrating acceptable performance in predicting adverse perinatal outcomes in our institutional setting [25].

Fetuses in both the 10th–20th percentile and >20th–<90th percentile groups were monitored according to institutional guidelines. Surveillance was individualized based on maternal comorbidities, clinical evolution, and sonographic findings. Follow-up typically included serial ultrasounds to assess fetal growth and amniotic fluid volume. Doppler evaluations were performed selectively in either group when additional risk factors or concerning findings (e.g., FGR suspicion or abnormal fluid levels) were identified, consistent with standard clinical practice during the study period. Prenatal follow-up and delivery management for all pregnancies were conducted at the discretion of the attending obstetrician, in accordance with institutional protocols and current clinical guidelines. The decisions regarding the timing of delivery were individualized, taking into account gestational age, fetal well-being, Doppler findings when performed, and maternal clinical factors.

The primary outcome was a composite adverse neonatal outcome (ANeO), defined as the presence of one or more of the following: umbilical artery pH ≤ 7.1, taken immediately after delivery; 5 min Apgar score ≤ 7 [26], which was calculated by trained and certified neonatologists at 1 min and 5 min after birth; admission to the neonatal intensive care unit (NICU) in the first 48 h of life, where the decision to admit to the NICU was made by the attending neonatologist; hypoglycemia (<45 mg/dL [2.5 mmol/L] in the first 24 h of life); non-reassuring fetal state (i.e., category II cardiotocographic monitoring that does not recover after intrauterine resuscitation maneuvers or category III) requiring an emergency delivery, defined according to the American College of Obstetricians and Gynecologists (ACOG) guidelines [27]; respiratory distress syndrome based on typical history, clinical manifestations, arterial blood gas analysis, and chest X-ray (CXR) findings [28]; intraventricular hemorrhage identified by imaging studies in the first 24 h of life; hypoxic–ischemic encephalopathy defined according to the following criteria: Apgar score: ≤5 at 10 min, umbilical cord blood gas analysis: pH < 7.00 and base deficit ≥ 12−16 mmol/L, and perinatal death after 28 weeks or within the first 7 days postpartum. Secondary outcomes included progression to FGR and low birth weight, as defined by the World Health Organization (WHO) criteria: [29] Extremely Low Birth Weight (ELBW): less than 1000 g, Very Low Birth Weight (VLBW): less than 1500 g (typically between 1000 and 1499 g), Low Birth Weight (LBW): less than 2500 g (typically between 1500 and 2499 g), Adequate/Normal Birth Weight: between 2500 and 3999 g, High Birth Weight: equal to or greater than 4000 g; oligohydramnios; delivery mode; and meconium-stained amniotic fluid.

Demographic data, medical history, fetal biometric parameters, and perinatal outcomes were retrieved from the electronic medical records. Importantly, any fetus initially assigned to a study group was analyzed within that original group, regardless of subsequent changes in EFW percentile. This approach was used to preserve the integrity of the analysis and maintain an intention-to-treat framework. This project was approved by the Institutional Research Ethics Committee (protocol #: CEI-RETRO-01-2024). The Strengthening the Reporting of Observational Studies in Epidemiology (STROBE) statement for the present study is reported as Appendix A.

### Statistical Analysis

The characteristics of the study population are presented as means (standard deviations [SDs]), medians (interquartile ranges [IQRs]), or n (%). The distribution of continuous variables was analyzed using histograms. Both groups were compared regarding baseline variables using standardized differences, an intuitive index for comparing baseline characteristics in clinical trials and observational studies. An absolute standardized difference greater than 0.20 would indicate an imbalance between the groups [30]. Ultrasound comparisons of follow-up fetal growth assessments and final diagnosis of fetal growth before delivery and birth weight were performed with nonparametric tests, with a Mann–Whitney U test for continuous variables and Fisher’s exact test for categorical variables.

Adverse perinatal and obstetric outcomes were compared according to the EFW percentile in the early third trimester [18], 10th–20th percentile vs. those with a >20th Hadlock percentile (but less than 90) [9,11,31]. For binary fetal, neonatal, and obstetric outcomes, modified Poisson regression models with robust error variances were used to calculate crude and adjusted risk ratios (RRs) and their corresponding 95% confidence intervals (CIs) to assess the presence and magnitude of a potential association between a 10th to 20th fetal growth centile and the composite adverse neonatal outcome. Continuous variables were log-transformed to report the effects as crude or adjusted ratios of geometric means (RGMs) calculated by linear regression [32]. The following covariates were included in the adjusted regression models: maternal age, gravidity, pregestational body mass index (BMI), history of preeclampsia, diabetes (both pregestational and gestational), passive tobacco exposure, antiphospholipid syndrome, systemic lupus erythematosus, and history of stillbirth. A complete case analysis was performed for all statistical analyses. Data were collected from electronic medical records and ultrasound reports by a researcher blinded to the outcomes of interest.

We performed an *a priori* sample size calculation using a two-sided significance level (α) of 0.05 and a power (1-β) of 80%, based on our primary outcome: ANeO. Institutional data indicated a baseline incidence of approximately 25% for this composite outcome among fetuses with an EFW > 20th–<90th percentile between 28.0 and 30.0 weeks of gestation. We aimed to detect an absolute risk difference of 10%—i.e., an expected increase to 35% in the 10th–20th percentile group—which we considered clinically meaningful. Based on these assumptions, a minimum of 329 participants per group (658 in total) was required. Statistical analyses were performed in Stata (version 19.0; StataCorp, College Station, TX, USA).

## 3. Results

A total of 650 pregnancies met the eligibility criteria and were included in the analysis: there were 323 fetuses in the 10th–20th percentile group and 327 in the >20th–<90th percentile control group. Although the initial *a priori* sample size calculation indicated a target of 329 participants per group, the final numbers were slightly lower due to strict inclusion criteria and data completeness requirements. Baseline characteristics are presented in Table 1. Patients in the >20th–<90th percentile group were older, had a higher pregestational weight, and had less exposure to tobacco. Otherwise, the groups were similar in terms of other characteristics (Table 1). Notably, 92.46% (N = 601) of the entire cohort was born at term.

Regarding fetal growth assessments (Table 2), the mean gestational age for the first growth scan in the entire cohort was 28.5 weeks (IQR: 28.0, 29.6), with an EFW of 1239.5 g (IQR: 1098, 1418). The median EFW centile in the first growth scan for the 10th–20th centile group was 15 (IQR: 12, 18) and 42 (IQR: 30, 58) for the >20th–<90th centile group (*p* < 0.001, Table 2). There were no differences in the gestational age for the last growth scan; however, the total number of scans performed was higher for the 10th–20th centile group. The latency between the last growth ultrasound performed and delivery was 13 days (IQR: 5–25) for the 10th–20th percentile group and 17 days (IQR: 9–27) for the control group (*p* < 0.001; Table 2).

Table 3 highlights the fetal growth trajectories from the first growth scan to the last ultrasound performed before delivery, as well as the birthweight classification. Notably, fetal growth restriction, which was diagnosed according to the international Delphi consensus criteria [21]; small for gestational age (i.e., EFW > 3rd—<10th centile with a normal Doppler assessment); and low birth weight according to WHO [30,31] occurred more frequently in the 10th–20th percentile group when compared to higher percentiles (Table 3). Oligohydramnios (aRR: 2.91, 95% CI: [1.17, 7.22], *p* = 0.021) and cesarean deliveries (aRR: 1.13, 95% CI: [1.02, 1.25], *p* = 0.019) were also more frequent in the 10th–20th percentile group. The median birth weight was 2736 g (IQR: 2455, 2985) in the 10th–20th percentile group and 3020 g (IQR: 2765, 3295) in the control group (aRGM: 0.90, 95% CI: [0.87, 0.92], *p* < 0.001). Moreover, meconium-stained amniotic fluid occurred more frequently in the 10th–20th percentile group compared to the >20th–<90th group (aRR: 4.89, 95% CI: [1.07, 22.27], *p* = 0.040, Table 4). ANeO occurred in 37.5% of the entire cohort. ANeO was present in 45.8% of the 10th-20th percentile group and 29.4% of the control group (aRR: 1.51, 95% CI: [1.22, 1.86], *p* < 0.001, Table 4). The comparisons of individual components for the composite outcome are shown in Table 4. There were no perinatal deaths.

## 4. Discussion

The results of this single-center retrospective cohort study suggest that in singleton pregnancies, fetuses that initiate the growth spurt stage in the early third trimester [17] at the 10th to 20th percentile have a higher risk of neonatal morbidity, fetal growth restriction, and low birth weight than fetuses with an EFW between the >20th and <90th percentile. Our decision to focus on fetuses with an EFW between the 10th and 20th percentiles was informed by the growing recognition that perinatal risk increases gradually across the weight distribution, without a strict inflection point at the 10th percentile. As highlighted by Ganzevoort et al. [9], outcomes such as perinatal mortality and NICU admission show a linear association with decreasing birthweight percentile. This reinforces the concept that fetuses in the 10th–20th percentile range may represent a subtly at-risk group, potentially overlooked under current surveillance paradigms. Our findings support the need to re-evaluate monitoring strategies for these cases and highlight the importance of future studies exploring more granular risk stratification within the traditionally defined normal range.

Previous reports [33,34,35] have shown that adverse perinatal outcomes are proportionally increased as the percentiles of EFW decrease during pregnancy. Moreover, growth assessments have been traditionally dichotomized as normal or abnormal based on certain thresholds (e.g., EFW or AC < 10th percentile). A retrospective cohort study by Vasak et al. [8] found that birth weights around the 80th–84th percentiles were associated with the lowest risk of adverse perinatal outcomes, including fetal or neonatal death within 7 days after delivery. While this finding offers an interesting perspective on the potential benefits of higher birthweight centiles, it should be interpreted with caution, as it stems from a single study and may not be generalizable across different populations or clinical settings. In contrast, most contemporary definitions of FGR—such as those proposed through the Delphi consensus process [21]—focus on identifying increased risk associated with lower fetal weight percentiles, particularly below the 10th percentile. Additionally, the conceptual model proposed by Ganzevoort et al. [9] supports a continuum of risk rather than a dichotomous threshold, showing that adverse outcomes increase progressively as estimated fetal weight declines—even within ranges traditionally considered normal. Taken together, these perspectives emphasize the importance of interpreting fetal size percentiles within a broader clinical context, rather than relying solely on fixed cutoffs. Khalil AA. and colleagues [36] reported that a low fetal cerebroplacental ratio at term increases the risk of operative delivery for fetal compromise and neonatal intensive care unit admission in small-for-gestational-age fetuses. Interestingly, they highlighted that their findings also apply to appropriate-for-gestational-age EFW, which supports the concept that even this population could be prone to a certain degree of placental insufficiency. Notably, the risk of adverse perinatal outcomes is a continuum for the EFW or AC percentiles, as highlighted in the findings of a population-based linkage study by Iliodromiti S. and colleagues [37] that suggest that early-term delivery or closer follow-up for pregnancies with predicted birth weights ≤ 25th percentile or ≥ 85th percentile might reduce adverse outcomes like stillbirth and admission to the NICU.

Our findings contribute to a growing body of evidence suggesting that the risk of adverse perinatal outcomes does not abruptly emerge below the 10th percentile but rather increases progressively as fetal size decreases. As shown in previous studies (e.g., Ganzevoort et al. [9]), a continuum of vulnerability appears across the fetal weight spectrum, with even “normal-but-low” percentile ranges (e.g., 10th–20th) associated with an elevated risk. In our cohort, nearly one-third of fetuses in this range were later classified as small for gestational age or diagnosed with fetal growth restriction, and the composite rate of adverse outcomes was significantly higher than in the reference group. These findings suggest that current surveillance thresholds may not fully capture at-risk fetuses in this intermediate range. However, further research is needed to validate these observations across diverse populations and healthcare settings before considering changes to clinical management.

### 4.1. Research Implications

Fetuses classified as “normal” according to their EFW [18] percentile could be prone to sentinel events during labor or placental dysfunction. Still, they could have a normal fetal weight, as fetal size only reflects the transplacental passage of nutrients [8]. Furthermore, there is no EFW percentile above which favorable outcomes can be assured. Notably, genetics play an essential role in fetal size. Some fetuses are predisposed to be smaller than others, even in the absence of any medical problems. Moreover, small-for-gestational-age fetuses are typically born healthy and do not experience long-term complications. In contrast, FGR may be present due to severe placental insufficiency, and this clinical profile has been extensively studied, mainly in the context of an EFW/AC percentile below 10 [38,39]. Our findings underscore the need for further research to better delineate the continuum of risk across the fetal weight distribution, particularly among fetuses traditionally classified as “normal” but located at the lower end of the percentile spectrum. While this study identified significantly higher rates of adverse outcomes in the 10th–20th percentile group, it was not powered to assess narrower percentile bands or determine an evidence-based cutoff for intensified surveillance. Future large-scale, prospective studies should aim to stratify fetuses using smaller percentile increments (e.g., 5% intervals) and include diverse populations and care settings to enhance generalizability. Importantly, future research should also evaluate the role of fetal surveillance intensity and delivery decision-making in mediating outcomes within these groups. Such efforts are essential before translating our findings into practice and could contribute to refining surveillance protocols and contemporary definitions of fetal growth restriction and “at-risk” fetal size thresholds. Furthermore, the economic impact and workforce implications of closer monitoring and follow-up in fetuses with an EFW percentile between 10 and 20 in the early third trimester warrant further study. Likewise, whether other biomarkers (e.g., placenta-enriched molecules and sFlt-1/PlGF ratios) [40,41] can aid in identifying a higher risk of fetal growth restriction, low birth weight, and neonatal morbidity in fetuses between the 10th and 20th EFW percentile throughout gestation merits further investigation. Future studies should investigate the impact of using alternative fetal growth standards, such as the WHO or INTERGROWTH-21st charts, to better understand how different definitions of fetal size influence neonatal outcomes. In addition, incorporating biometric data from earlier gestational ages, including second-trimester ultrasounds, would allow for individualized assessments of fetal growth trajectories. Longitudinal evaluation of growth velocity may help distinguish constitutionally small fetuses from those with emerging growth restriction, offering a more nuanced approach to risk stratification beyond a single-point EFW/AC measurement. Further research should also include fetuses with an EFW/AC below the 10th percentile—traditionally considered the highest-risk group—as a reference population. This would provide a more comprehensive framework for interpreting the relative risk associated with fetuses in the 10th–20th percentile range and clarify whether this subgroup represents a distinct clinical entity or a point along a continuous risk spectrum. Lastly, future studies should investigate the placental pathology of this intermediate group, particularly focusing on lesions such as maternal or fetal vascular malperfusion, which may provide mechanistic insights into adverse outcomes and guide more targeted clinical management.

### 4.2. Strengths and Limitations

Our study has some strengths. First, this cohort study included a population of fetuses in which maternal–fetal medicine specialists performed the scans according to standardized guidelines [1,2]. In addition, in those fetuses that presented growth alterations such as FGR or SGA during follow-up, the diagnosis, follow-up, and delivery mode adhered to current practice and clinical recommendations [21,39]. Furthermore, maternal and pregnancy clinical features potentially affecting the primary outcome were similar between groups and were considered in the statistical analysis. However, we recognize that our study has some limitations. As a single-center retrospective cohort study, the results may be subject to intervention bias, since perinatal outcomes could have been influenced by local follow-up protocols, management decisions, and the attending obstetricians’ discretion. The high overall cesarean section rate likely reflects local practice patterns and decision-making thresholds. This limits the external generalizability of our findings. Furthermore, our institutional protocol includes a routine third-trimester growth scan for all pregnancies, which may not reflect standard practice in settings where ultrasound use is more selective. These local practices, along with regional differences in referral patterns, intervention thresholds, and clinical decision-making, should be considered when interpreting the applicability of our findings to broader populations. One additional limitation of this study is the slight shortfall in achieving the exact target sample size, with 323 and 327 participants included in the two groups instead of the 329 per group calculated in the *a priori* power analysis. This was primarily due to the application of strict eligibility criteria and the exclusion of cases with incomplete data. While this may have resulted in a marginal reduction in statistical power, a post hoc power analysis based on the observed difference in the primary outcome (45.8% vs. 29.4%) demonstrated that the study retained over 95% power (i.e., 0.99) to detect the effect, supporting the robustness of the findings. Like all observational studies, residual confounding could affect the reported effects [42]. Finally, the individual components of the composite outcome were relatively rare, and the study was neither designed nor powered to assess these events; therefore, the reported effects for these outcomes should be interpreted with caution.

## 5. Conclusions

In this single-center cohort study, fetuses entering the accelerated growth stage between the 10th and 20th percentile of EFW in the early third trimester had a higher risk of neonatal morbidity than those entering this stage at a higher percentile. Furthermore, the proportion of fetuses diagnosed later with fetal growth restriction and low birth weight was greater in the 10th and 20th percentile group. Further research in different contexts and larger samples is needed to confirm our findings and to explore the potential clinical utility of a closer follow-up in fetuses with an EFW between the 10th and 20th percentile.

## Figures and Tables

**Figure 1 diagnostics-15-02251-f001:**
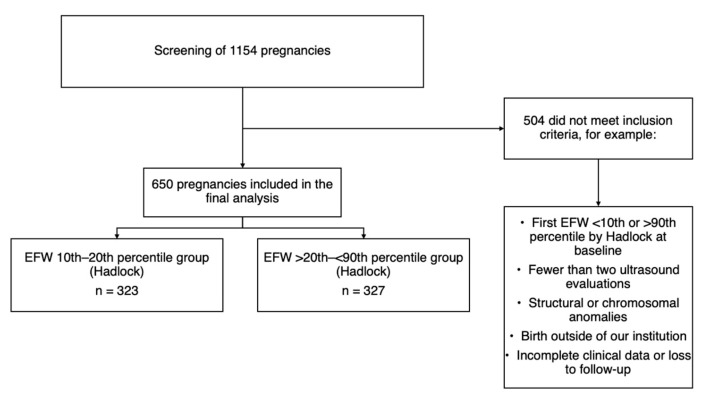
Flowchart that outlines the study population.

**Figure 2 diagnostics-15-02251-f002:**
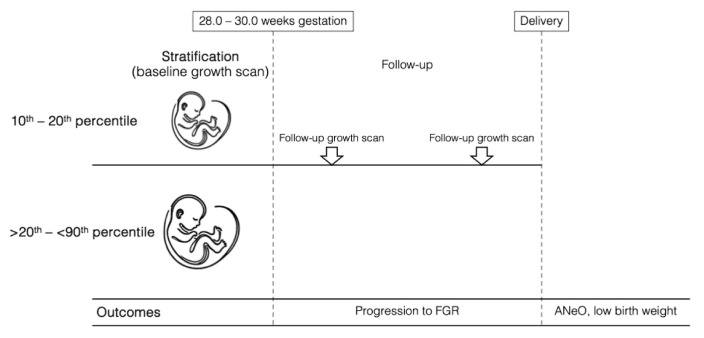
Study design and primary outcomes of interest.

**Table 1 diagnostics-15-02251-t001:** Patient characteristics.

	Full Cohort	10th–20th Centile	>20th–<90th Centile	Standardized Difference ^a^
	N = 650	N = 323	N = 327	
Age, years, median (IQR)	29 [24, 34]	28 [24, 33]	30 [25, 35]	−0.251
Gravidity, median (IQR)	2 [1, 3]	2 [1, 3]	2 [2, 3]	−0.245
Pregestational weight, kg, median (IQR)	69.7 [60, 80]	68 [57, 79]	71 [63, 82]	−0.310
Pregestational BMI, kg/m^2^, median (IQR)	27.9 [24.6, 31.6]	27.8 [23.8, 31.3]	28.3 [25.2, 32.4]	−0.213
Hypertensive disorders of pregnancy, n (%)				0.101
None	579 (89.1)	276 (85.4)	303 (92.7)	
Preeclampsia	33 (5.1)	28 (8.7)	5 (1.5)	
Gestational hypertension	--	--	--	
Chronic hypertension	38 (5.8)	19 (5.9)	19 (5.8)	
History of preeclampsia, n (%)	74 (11.4)	36 (11.1)	38 (11.6)	−0.014
Diabetes, n (%)				−0.070
None	610 (93.8)	305 (94.5)	305 (93.3)	
Pregestational	5 (0.8)	4 (1.2)	1 (0.3)	
Gestational	35 (5.4)	14 (4.3)	21 (6.4)	
Tobacco exposure, n (%)	34 (5.2)	25 (7.7)	9 (2.8)	0.224
APS or SLE, n (%)	22 (3.4)	16 (5.0)	6 (1.8)	0.172
History of stillbirth, n (%)	8 (1.2)	4 (1.2)	4 (1.2)	0.001

IQR = Interquartile range; BMI = Body mass index; APS = Antiphospholipid syndrome; SLE = Systemic lupus erythematosus. ^a^ = Standardized difference for 10th–20th centile vs. >20th to <90th centile.

**Table 2 diagnostics-15-02251-t002:** Fetal growth assessments.

	Full Cohort	10th–20th Centile	>20th–<90th Centile	*p*-Value
	N = 650	N = 323	N = 327	
Gestational age, first growth scan, median (IQR)	28.5 [28.0, 29.6]	28.4 [27.6, 29.5]	28.5 [28.1, 30.0]	0.005
EFW, first growth scan, median (IQR)	1239.5 [1098, 1418]	1132 [1036, 1288]	1332 [1213, 1531]	<0.001
EFW centile, first growth scan, median (IQR)	21 [15, 43]	15 [12, 18]	42 [30, 58]	<0.001
Gestational age, last growth scan, median (IQR)	36 [34.3, 36.6]	36.0 [34.1, 36.6]	36.0 [34.4, 36.6]	0.760
EFW, last growth scan, median (IQR)	2457 [2221, 2708]	2355 [2047, 2579]	2606 [2350, 2823]	<0.001
EFW centile, last growth scan, median (IQR)	25 [11, 43]	16.0 [7.0, 26.6]	37.0 [21.0, 51.0]	<0.001
Total number of growth scans performed, median (IQR)	3 [2, 4]	3 [2, 4]	3 [2, 3]	0.001
Latency between last growth scan and delivery, days, median (IQR)	15 [7, 26]	13 [5, 25]	17 [9, 27]	<0.001

IQR = Interquartile range; EFW = Estimated fetal weight. Note: Given the large sample size, small differences between groups—such as in gestational age at first growth scan—may reach statistical significance (e.g., *p* = 0.005) despite being clinically negligible.

**Table 3 diagnostics-15-02251-t003:** Final growth diagnosis before and after delivery.

	Full Cohort	10th–20th Centile	>20th–<90th Centile	*p*-Value
	N = 650	N = 323	N = 327	
Final fetal growth diagnosis in the last ultrasound before delivery, n (%)				<0.001
Fetal growth restriction	89 (13.7)	68 (21.1)	21 (6.4)	
Small for gestational age	70 (10.8)	53 (16.4)	17 (5.2)	
10th–20th centile	121 (18.6)	86 (26.6)	35 (10.7)	
>20th centile	370 (56.9)	116 (35.9)	254 (77.7)	
Gestational age at fetal growth restriction diagnosis, weeks, median (IQR) ^a^	35.4 [33.0, 37.0]	35.0 [32.5, 36.5]	36.4 [35.0, 37.0]	0.043
Gestational age at small for gestational age at diagnosis, weeks, median (IQR) ^a^	36.0 [34.6, 36.5]	35.4 [34.2, 36.1]	36.6 [36.1, 37.0]	<0.001
Birth weight classification ^b^, n (%)				<0.001
Extremely low	1 (0.2)	1 (0.3)	0 (0.0)	
Very low	--	--	--	
Low	108 (16.6)	89 (27.6)	19 (5.8)	
Adequate	538 (82.7)	233 (72.1)	305 (93.3)	
High	3 (0.5)	0 (0.0)	3 (0.9)	

^a^ = If applicable. ^b^ = First weight recorded after birth classified according to the WHO criteria mentioned above [30,31].

**Table 4 diagnostics-15-02251-t004:** Fetal, neonatal, and obstetric outcomes.

	Full Cohort	10th–20th Centile	>20th–<90th Centile	Crude Effect (95% CI)	Adjusted Effect (95% CI) ^a^	*p*-Value ^a^
	N = 650	N = 323	N = 327			
Oligohydramnios, n (%)	28 (4.3)	22 (6.8)	6 (1.8)	3.71 (1.52, 9.04) ^b^	2.91 (1.17, 7.22) ^b^	0.021
Gestational age at the time of delivery, weeks, median (IQR)	38.2 [37.4, 39.1]	38.1 [37.2, 39.1]	38.3 [37.6, 39.1]	0.98 (0.98, 0.99) ^c^	0.99 (0.98, 0.99) ^c^	0.001
Induction of labor, n (%), indications (not mutually exclusive)	83 (15.7)	42 (15.2)	41 (16.4)	0.92 (0.62, 1.37) ^b^	0.86 (0.57, 1.30) ^b^	0.494
Term pregnancy (i.e., 40 weeks’ gestation)	45 (8.8)	19 (7.1)	26 (10.7)	--	--	
Hypertensive disorders of pregnancy	20 (3.9)	9 (3.4)	11 (4.5)	--	--	
Pregestational or gestational diabetes	0 (0)	0 (0)	0 (0)	--	--	
Premature rupture of membranes	11 (2.2)	7 (2.6)	4 (1.6)	--	--	
Fetal growth restriction	8 (1.6)	6 (2.2)	2 (0.8)	--	--	
Oligohydramnios	5 (1.0)	4 (1.5)	1 (0.4)	--	--	
Delivery mode, n (%)						
Cesarean delivery	456 (70.2)	240 (74.3)	216 (66.1)	1.12 (1.01, 1.24) ^b^	1.13 (1.02, 1.25) ^b^	0.019
Birth weight, grams, median (IQR)	2862.5 [2595, 3146]	2736 [2455, 2985]	3020 [2765, 3295]	0.88 (0.86, 0.91) ^c^	0.90 (0.87, 0.92) ^c^	<0.001
Five-minute Apgar, median (IQR)	9 [9, 9]	9 [9, 9]	9 [9, 9]	0.99 (0.99, 1.00) ^c^	0.99 (0.99, 1.00) ^c^	0.150
Meconium-stained amniotic fluid, n (%)	15 (2.3)	13 (4.0)	2 (0.6)	6.58 (1.49, 28.96) ^b^	4.89 (1.07, 22.27) ^b^	0.040
Neonatal jaundice, n (%)	203 (31.2)	115 (35.6)	88 (26.9)	1.32 (1.04, 1.66) ^b^	1.33 (1.05, 1.69) ^b^	0.016
Necrotizing enterocolitis, n (%)	3 (0.5)	2 (0.6)	1 (0.3)	2.02 (0.18, 22.26) ^b^	1.60 (0.12, 19.79) ^b^	0.714
Neonatal sepsis, n (%)						
Early	3 (0.5)	2 (0.6)	1 (0.3)	2.02 (0.18, 22.26) ^b^	25.87 (5.63, 118.76) ^b^	<0.001
Late	1 (0.2)	0 (0.0)	1 (0.3)	--	--	
Mechanical ventilation, n (%)	11 (1.7)	5 (1.5)	6 (1.8)	0.84 (0.25, 2.73) ^b^	1.02 (0.31, 3.38) ^b^	0.964
Continuous positive airway pressure, n (%)	37 (5.7)	20 (6.2)	17 (5.2)	1.19 (0.63, 2.23) ^b^	1.13 (0.61, 2.09) ^b^	0.681
Umbilical cord pH ≤ 7.1, n (%)	11 (1.7)	5 (1.5)	6 (1.8)	0.84 (0.25, 2.73) ^b^	0.79 (0.25, 2.46) ^b^	0.688
Five-minute Apgar ≤ 7, n (%)	5 (0.8)	3 (0.9)	2 (0.6)	1.51 (0.25, 9.04) ^b^	1.72 (0.34, 8.62) ^b^	0.505
NICU admission, n (%)	146 (22.5)	99 (30.7)	47 (14.4)	2.13 (1.56, 2.91) ^b^	2.06 (1.50, 2.84) ^b^	<0.001
Hypoglycemia, n (%)	24 (3.7)	15 (4.6)	9 (2.8)	1.68 (0.74, 3.80) ^b^	1.51 (0.65, 3.52) ^b^	0.332
Emergency C-section due to non-reassuring fetal state, n (%)	101 (15.5)	64 (19.8)	37 (11.3)	1,75 (1,20, 2,54) ^b^	1.56 (1.05, 2.31) ^b^	0.026
Respiratory distress syndrome, n (%)	115 (17.7)	67 (20.7)	48 (14.7)	1.41 (1.00, 1.98) ^b^	1.41 (0.99, 1.99) ^b^	0.050
Intraventricular hemorrhage, n (%)	5 (0.8)	3 (0.9)	2 (0.6)	1.51 (0.25, 9.04) ^b^	1.52 (0.17, 13.67) ^b^	0.705
Hypoxic ischemic encephalopathy, n (%)	5 (0.8)	1 (0.3)	4 (1.2)	0.25 (0.02, 2.25) ^b^	0.29 (0.02, 2.97) ^b^	0.300
Perinatal death, n (%)	0 (0)	0 (0.0)	0 (0.0)	--	--	
Composite outcome, n (%) ^d^	244 (37.5)	148 (45.8)	96 (29.4)	1.56 (1.27, 1.91) ^b^	1.51 (1.22, 1.86) ^b^	<0.001

RR = Risk ratio; CI = Confidence interval; IQR = Interquartile range; NICU = Neonatal intensive care unit. ^a^ = Effect adjusted for maternal age, gravidity, pregestational body mass index, history of preeclampsia, diabetes (pregestational and gestational), tobacco exposure, antiphospholipid syndrome, systemic lupus erythematosus, and history of stillbirth. ^b^ = Risk ratio. ^c^ = Ratio of geometric means. ^d^ = Defined as umbilical artery pH ≤ 7.1, 5 min Apgar score ≤ 7, admission to the neonatal intensive care unit in the first 48 h of life, hypoglycemia (<45 mg/dL [2.5 mmol/L] in the first 24 h of life), non-reassuring fetal state (i.e., category II cardiotocographic monitoring that does not recover after intrauterine resuscitation maneuvers or category III) requiring an emergency birth, respiratory distress syndrome, intraventricular hemorrhage, hypoxic–ischemic encephalopathy, and perinatal death.

## Data Availability

The datasets generated and analyzed during the current study are available from the corresponding author upon reasonable request.

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
