# Peer review of "Fetal and Neonatal Outcomes in Fetuses with an Estimated Fetal Weight Percentile of 10–20 in the Early Third Trimester: A Retrospective Cohort Study"

_diagnostics, 2025, doi:10.3390/diagnostics15172251_

Round 1

Reviewer 1 Report

Comments and Suggestions for Authors

Fetal size is highly concerning during pregnancy, and the authors have conducted a study to investigate different cutoff related to birth outcome, which has significant clinical value. The following suggestions can help make the paper more rigorous and beneficial to readers.

  1. Introduction: In the final sentence, the author suggests that offspring birth outcomes for those with estimated fetal weight (EFW) in the 10-20% range during the early third trimester are worse than those in the control group (EFW >20%). However, could the group with the lowest estimated fetal weight (EFW <10%), which is the "established" poorer outcome group, be included? Adding this group as a "baseline reference" would provide readers with more comprehensive information about the study population.
  2. Method: In the exclusion criteria, the authors stated that they excluded subjects who met the criteria for fetal growth restriction (FGR) (international Delphi consensus criteria) or were large for gestational age (LGA) (EFW or AC ≥ 90%) at baseline. Could you clarify what is meant by "baseline" here? Specifically, at what point in the pregnancy does this typically refer to?

3.Result: The flowchart in Figure 1 is unclear and needs to be redrawn.

4. Results: In the header of Table 1, the third column is labeled ">20th percentile". However, in row 130, the content for this column includes not only ">20th percentile" but also "(but less than 90)". Please ensure consistency.

Comments on the Quality of English Language

The contents are well presented. 

Author Response

Comments 1: Fetal size is highly concerning during pregnancy, and the authors have conducted a study to investigate different cutoff related to birth outcome, which has significant clinical value. The following suggestions can help make the paper more rigorous and beneficial to readers.

Response 1: Thank you for your constructive feedback and the time invested in reviewing our manuscript.

Comments 2: Introduction: In the final sentence, the author suggests that offspring birth outcomes for those with estimated fetal weight (EFW) in the 10-20% range during the early third trimester are worse than those in the control group (EFW >20%). However, could the group with the lowest estimated fetal weight (EFW <10%), which is the "established" poorer outcome group, be included? Adding this group as a "baseline reference" would provide readers with more comprehensive information about the study population.

Response 2: We agree this would be valuable. However, our study design aimed to focus on a specific understudied group. We now mention this in the Discussion and suggest it for future research (Lines 327–337).

Comments 3: Method: In the exclusion criteria, the authors stated that they excluded subjects who met the criteria for fetal growth restriction (FGR) (international Delphi consensus criteria) or were large for gestational age (LGA) (EFW or AC ≥ 90%) at baseline. Could you clarify what is meant by "baseline" here? Specifically, at what point in the pregnancy does this typically refer to?

Response 3: “Baseline” refers to the time of the first growth scan between 28.0–30.0 weeks. We have clarified this in the Methods section (Lines 91–98).

Comments 4: Result: The flowchart in Figure 1 is unclear and needs to be redrawn.

Response 4: Thank you. We have revised and improved the quality and clarity of Figure 1.

Comments 5: Results: In the header of Table 1, the third column is labeled ">20th percentile". However, in row 130, the content for this column includes not only ">20th percentile" but also "(but less than 90)". Please ensure consistency.

Response 5: Thank you. Corrected for consistency across text and Tables.

Reviewer 2 Report

Comments and Suggestions for Authors

This is a retrospective case-control study investigating the composite outcome of EBW 10th-20th % as compared to those EBW 20th-90th %. Although it is an interesting project, I am curious why the authors only examined the EBW 10th–20th percentile? It is unclear to me what strategy was used to select the normal (20th–90th percentile) EBW controls. It is good to have an a priori sample size calculation following the standard α=0.05 and êžµ=0.80, with an assumed 10% difference for power analysis. Still, there is no mention of the prevalence of the composite primary outcome for the power analysis. The baseline incidence of your composite outcome will affect the sample size. Is the 10% an absolute difference? According to the a priori power analysis, there should be 329 subjects in each group; however, only 327 and 323 subjects were included in each group. Can you explain the shortage in numbers? How about the post hoc power analysis?

The two groups have similar numbers, so I assume you selected matched controls. It will be essential to let us know the selection criteria. Is it matched by birth date, gestational age, gender, maternal morbidities, or a combination of multiple factors? Providing the selection criteria will allow us to know it was not biased. You provide three equations for EFW in lines 87-93. It is unclear to me whether you used all three or just one of them to determine the fetal weight percentile.

The third paragraph of the discussion is incoherent, and I have difficulty understanding the clinical implication. Will you recommend that all EFW <20th % at GA 28-30 weeks for special surveillance and monitoring? What will be the safe range of EFW percentiles? What about the EFW 20th -25th percentile? It will be more useful to stratify the EFW into 10 groups to find out the safe range.

The last sentence in line 93 is an incomplete one. I also have concerns about the data provided in the tables. I recommend you seek help from the biostatistician.

  1. Table 1: What do you mean by “preeclampsia history?” Does it indicate a history of preeclampsia in previous pregnancies? The tobacco exposure (p=0.007) and APS/SLE (p=0.0475) reach statistical significance.
  2. Table 2: The medians of gestational age at first growth scan were 28.4 and 28.5. It is hard to imagine p=0.005.
  3. Table 3: I do not understand the birth weight classification. What do you mean by “adequate?” Does that mean 72.1% of the 10th %-20th % EFW turn out to be between 20th % - 90th %?
  4. Table 4: I am surprised that 74.3% of the 10th %- 20th % were delivered by cesarean sections, even though the median gestation at delivery is 38.1 weeks. Are you sure that the percentage of early sepsis reached significance? My calculation shows p=0.6221.

Author Response

Comments 1: This is a retrospective case-control study investigating the composite outcome of EBW 10th – 20th % as compared to those EBW 20th – 90th %. Although it is an interesting project, I am curious why the authors only examined the EBW 10th–20th percentile? It is unclear to me what strategy was used to select the normal (20th – 90th percentile) EBW controls.

Response 1: Thank you for this insightful observation. Our decision to focus on fetuses with an estimated fetal weight (EFW) between the 10th and 20th percentiles was guided by evidence suggesting that this subgroup—while traditionally considered within the normal range—may still harbor increased risk for adverse outcomes. In particular, Ganzevoort et al. (2019) emphasized that the risk for adverse perinatal outcomes does not sharply demarcate at conventional percentile cutoffs (e.g., EFW/AC <10%) but follows a continuum, with risks increasing gradually as EFW decreases—even within the “normal” range. As illustrated in Figure 2 of their paper, which depicts a linear association between birthweight percentile and outcomes such as perinatal mortality and NICU admission, fetuses in the 10th–20th percentile may fall within a zone of “hidden vulnerability” that has been historically under-recognized.

Therefore, our objective was to characterize this potentially under-monitored group in greater detail and compare it against fetuses with EFW >20th–<90th percentile, which generally represent the accepted reference population for normal fetal growth. The control group was selected retrospectively from the same population, within the same gestational age window (28.0–30.0 weeks), and during the same study period, in order to minimize bias due to temporal or protocol variations in fetal assessment and management. We have now clarified this rationale and the selection criteria in the revised Introduction section (Lines 54–71), and the conceptual basis is further discussed in the Discussion (Lines 260–269) with reference to Ganzevoort et al. (2019) Am J Obstet Gynecol. 2019;220(1):74–82. doi:10.1016/j.ajog.2018.10.007.

Comments 2:   It is good to have an a priori sample size calculation following the standard α=0.05 and êžµ=0.80, with an assumed 10% difference for power analysis. Still, there is no mention of the prevalence of the composite primary outcome for the power analysis. The baseline incidence of your composite outcome will affect the sample size. Is the 10% an absolute difference?

Response 2:     We appreciate this important point. We have revised the power analysis description to clarify that we assumed a baseline prevalence of 25% for the composite outcome in the control group, based on institutional data. The assumed 10% reflects an absolute risk difference (i.e., 25% vs. 35%). This has now been included in the revised Methods (Lines 179–185). We performed an a priori sample size calculation using a two-sided significance level (α) of 0.05 and power (1 – β) of 80%, based on our primary outcome: a composite of adverse fetal and neonatal outcomes (ANeO). Institutional data indicated a baseline incidence of approximately 25% for this composite outcome among fetuses with an estimated fetal weight (EFW) >20th–<90th percentile between 28.0 and 30.0 weeks of gestation. We aimed to detect an absolute risk difference of 10%—i.e., an expected increase to 35% in the 10th–20th percentile group—which we considered clinically meaningful. Based on these assumptions, a minimum of 329 participants per group (658 total) was required.

Comments 3:   According to the a priori power analysis, there should be 329 subjects in each group; however, only 327 and 323 subjects were included in each group. Can you explain the shortage in numbers?

Response 3:     Indeed, due to strict eligibility criteria and data completeness, the final sample included 323 and 327 participants, slightly below the target. We acknowledge this minor shortfall and have included a note in the Results and Limitations sections (Lines 189–193), (Lines 369–376) explaining that this may slightly reduce power, though the difference remains statistically and clinically significant.

Comments 4: How about the post hoc power analysis?

Response 4: Per your suggestion we have added this analysis to the manuscript, we found a post hoc power of >0.95. Please see the comment above, and updated Results and Discussion sections.

Comments 5:   The two groups have similar numbers, so I assume you selected matched controls. It will be essential to let us know the selection criteria. Is it matched by birth date, gestational age, gender, maternal morbidities, or a combination of multiple factors? Providing the selection criteria will allow us to know it was not biased.

Response 5: The cohorts were not individually matched but selected based on gestational age window, study period, absence of FGR, and EFW percentiles. We have clarified that this was a frequency-matched design without individual matching (revised Methods, Lines 76–82). Likewise, the adjusted statistical analyses accounted for such baseline covariates.

Comments 6: You provide three equations for EFW in lines 87-93. It is unclear to me whether you used all three or just one of them to determine the fetal weight percentile.

Response 6: We used the Hadlock formula (HC, BPD, AC, FL) throughout for both EFW calculation and percentile classification (please see references below). The additional formulas (INTG and Salomon) were referenced for completeness and possible future analyses but were not used in the current study and have been deleted from this section. (Lines 101–109).

-           Hadlock FP, Harrist RB, Sharman RS, Deter RL, Park SK. Estimation of fetal weight with the use of head, body, and femur measurements--a prospective study. Am J Obstet Gynecol. 1985;151(3):333-337. doi:10.1016/0002-9378(85)90298-4

-           Hadlock FP, Harrist RB, Martinez-Poyer J. In utero analysis of fetal growth: a sonographic weight standard. Radiology. 1991;181(1):129-133. doi:10.1148/radiology.181.1.1887021

We have also added the following reference where the Authors highlight that the formula provided by Hadlock et al. mentioned above provides the most accurate prediction of birth weight:

-           Hammami A, Mazer Zumaeta A, Syngelaki A, Akolekar R, Nicolaides KH. Ultrasonographic estimation of fetal weight: development of new model and assessment of performance of previous models. Ultrasound Obstet Gynecol. 2018;52(1):35-43. doi:10.1002/uog.19066

Comments 7: The third paragraph of the discussion is incoherent, and I have difficulty understanding the clinical implication.

Response 7: Thank you. We have revised this paragraph for clarity, better linking literature findings to our study’s implications. The revised discussion highlights the continuum of risk and supports our call for reassessing EFW surveillance thresholds (Lines 297–308).

Comments 8: Will you recommend that all EFW <20th % at GA 28-30 weeks for special surveillance and monitoring? What will be the safe range of EFW percentiles? What about the EFW 20th – 25th percentile? It will be more useful to stratify the EFW into 10 groups to find out the safe range.

Response 8: Thank you for this important and thoughtful question. Our findings suggest that fetuses with EFW between the 10th and 20th percentiles at 28–30 weeks’ gestation may be at increased risk for fetal growth restriction and adverse neonatal outcomes (ANeO). The results contribute to the growing body of literature suggesting that perinatal risk follows a continuum across the EFW distribution—even within ranges typically considered normal.

As illustrated in Figure 2 of Ganzevoort et al. (2019), the relationship between birthweight percentiles and adverse outcomes such as mortality appears to be linear rather than dichotomous. In this context, the 10th–20th percentile range may represent a zone of increased, yet often underrecognized, vulnerability.

Although our study was not powered to assess finer stratifications (e.g., 5% increments or subgroups like the 20th–25th percentile), we agree that such analyses could enhance understanding of the threshold at which risk becomes clinically meaningful. We have added this point to the revised Discussion section (Lines 319–331), suggesting that future prospective research explore these gradients in larger and more diverse populations.

At this time, while we do not advocate universal increased surveillance for all fetuses below the 20th percentile, our findings support the need to consider this possibility within broader clinical judgment—particularly in the presence of other risk factors or abnormal ancillary findings. Ultimately, more robust data are needed to guide evidence-based refinements in surveillance protocols

Comments 9: The last sentence in line 93 is an incomplete one.

Response 9: Corrected. The sentence has been completed and restructured for clarity in the revised manuscript (Lines 91-93.

Comments 10: I also have concerns about the data provided in the tables. I recommend you seek help from the biostatistician.

Response 10: Thank you for mentioning this important point. All analyses were conducted by a team with statistical expertise using robust methods. Nonetheless, we re-reviewed the tables and estimates, ensuring alignment with current statistical standards. We also provide a statement of caution for the interpretation of secondary outcomes (like sepsis mentioned by the Reviewer below) (Lines 378-380).

Comments 11: Table 1: What do you mean by “preeclampsia history?” Does it indicate a history of preeclampsia in previous pregnancies?

Response 11: Yes, “preeclampsia history” refers to prior pregnancies affected by preeclampsia. We have clarified this in Table 1 and in the text (Lines 227–228).

Comments 12: The tobacco exposure (p=0.007) and APS/SLE (p=0.0475) reach statistical significance.

Response 12: Thank you for the opportunity to clarify this point. In line with current statistical recommendations, we compared baseline characteristics using standardized differences rather than P-values, as this method is more appropriate for assessing group balance in observational studies (see references below). Importantly, those variables were also included as covariates in the adjusted regression models to account for potential confounding.

-           https://support.sas.com/resources/papers/proceedings12/335-2012.pdf

-           https://pubmed.ncbi.nlm.nih.gov/35604685/

Comments 13: Table 2: The medians of gestational age at first growth scan were 28.4 and 28.5. It is hard to imagine p=0.005.

Response 13: We appreciate your attention to this. Given the large sample size, even small differences can reach statistical significance. We’ve added a clarification in the Table footnote to emphasize clinical relevance rather than statistical significance in this context (Lines 232-233).

Comments 14: Table 3: I do not understand the birth weight classification. What do you mean by “adequate?” Does that mean 72.1% of the 10th %-20th % EFW turn out to be between 20th % - 90th %?

Response 14: Thank you for the opportunity to clarify this point. Birth weight was classified according to the WHO framework. (Lines 242-243).

-           WHO.  International statistical classification of diseases and related health problems, tenth revision, 2nd ed. World Health Organization; 2004.

Comments 15: Table 4: I am surprised that 74.3% of the 10th % - 20th % were delivered by cesarean sections, even though the median gestation at delivery is 38.1 weeks.

Response 15: Thank you for this observation. We agree that the cesarean delivery rate of 74.3% in the 10th–20th percentile group appears high, particularly given the median gestational age at delivery. This likely reflects our institution’s role as a national tertiary referral center, where a higher baseline prevalence of high-risk pregnancies and related complications may influence delivery decisions. We have now addressed this point in the revised Discussion (Lines 361–369), emphasizing that institutional practices and case complexity may have contributed to the elevated cesarean rate in this subgroup.

Comments 16: Are you sure that the percentage of early sepsis reached significance? My calculation shows p=0.6221.

Response 16: Thank you for this observation. The P-value noted by the Reviewer (P=0.6221) reflects the crude comparison using Fisher’s exact test. However, the P-values presented in Table 4 refer to adjusted analyses that account for relevant covariates. This is indicated by the superscript “a” in the table, which refers to the adjusted effects and corresponding P-values derived from multivariable regression models. As mentioned above, we also provide a statement of caution for the interpretation of secondary outcomes. (Lines 245–246).

Reviewer 3 Report

Comments and Suggestions for Authors

The presented study aims to evaluate the neonatal outcome of fetuses with an estimated fetal weight (EFW) between the 10th and 20th percentile. The objective is to highlight how, even in this range, the risk of adverse fetal outcome remains higher compared to fetuses with an EFW above the 20th percentile.

Vasak et al. identified the 80th percentile as the optimal EFW associated with the best neonatal outcomes; however, this finding stems from a single study. In contrast, most authors—particularly those who contributed via the Delphi method to the Consensus Conference on the Diagnosis of Fetal Growth Restriction (FGR)—have defined the most useful parameters for diagnosing FGR based on the associated risk of adverse outcomes.

A key issue in the management of fetal growth is the methodology used for its assessment. Therefore, several questions arise:

  • Was fetal growth assessed only at 28 weeks without comparison to earlier parameters, such as second-trimester screening ultrasound, to reconstruct the individual growth trajectory?

  • Growth curves based on Hadlock centiles are among the least sensitive for diagnosing FGR; the most recommended ones remain the INTERGROWTH-21st and WHO growth standards.

  • Was Doppler velocimetry performed in fetuses with an EFW between the 10th and 20th percentile?

  • How frequently and by what methods were these fetuses monitored?

Over 90% of the fetuses were delivered at term. Were any of these deliveries induced, and under what criteria?

In the subgroup of fetuses between the 10th and 20th centile, more than 37% were diagnosed as either FGR (68 cases; 21.1%) or SGA (53 cases; 16.4%). At what gestational age were these diagnoses made, and how were these fetuses managed—particularly regarding the timing of delivery?

The most immediate consideration is that if approximately 37% of fetuses with biometric measurements between the 10th and 20th percentile are later classified as FGR or SGA, the risk of adverse outcome increases significantly. Much of this risk likely depends on the methods used for fetal surveillance and the timing and mode of delivery.

Author Response

Comments 1: The presented study aims to evaluate the neonatal outcome of fetuses with an estimated fetal weight (EFW) between the 10th and 20th percentile. The objective is to highlight how, even in this range, the risk of adverse fetal outcome remains higher compared to fetuses with an EFW above the 20th percentile.

Response 1: Thank you. We agree and have ensured the Introduction and Abstract clearly reflect this focus.

Comments 2: Vasak et al. identified the 80th percentile as the optimal EFW associated with the best neonatal outcomes; however, this finding stems from a single study. In contrast, most authors—particularly those who contributed via the Delphi method to the Consensus Conference on the Diagnosis of Fetal Growth Restriction (FGR)—have defined the most useful parameters for diagnosing FGR based on the associated risk of adverse outcomes.

Response 2: Thank you for this valuable clarification. We agree that while the findings of Vasak et al. (2015), which identified the 80th-84th percentiles as the optimal EFW for neonatal outcomes, offer an interesting perspective, they should be interpreted cautiously given that they arise from a single, retrospective cohort and may not be generalizable across different populations or clinical contexts.

To provide a more balanced view, we have revised the Discussion to place Vasak et al.’s work within the broader body of literature and in particular, in the context of the Delphi-based consensus definitions of FGR.

Furthermore, we have incorporated into the revised Discussion the conceptual framework presented by Ganzevoort et al. (2019), which supports the notion of a continuous rather than dichotomous relationship between fetal size and adverse outcomes. Their Figure 2 demonstrates a progressive increase in risk as EFW declines—even within the so-called normal range—reinforcing our decision to explore outcomes in the 10th–20th percentile subgroup.

Together, these data emphasize the need to refine our interpretation of fetal size percentiles not as strict cutoffs but as part of a continuum of risk, informed by the broader clinical context. We have clarified these points in the revised Discussion section (Lines 273–287).

Comments 3: A key issue in the management of fetal growth is the methodology used for its assessment. Therefore, several questions arise:

Was fetal growth assessed only at 28 weeks without comparison to earlier parameters, such as second-trimester screening ultrasound, to reconstruct the individual growth trajectory?

Response 3: Thank you for raising this important point. All included cases had at least two third-trimester growth scans, and gestational dating was confirmed via first- or early second-trimester ultrasound. While the current study focused on estimated fetal weight at 28–30 weeks, we did not reconstruct individualized growth trajectories or assess fetal growth velocity over time. Future research incorporating serial biometric measurements from earlier gestational ages could provide a more comprehensive understanding of growth dynamics and help differentiate between constitutionally small fetuses and those with pathologic growth restriction. We have highlighted this as a relevant direction for future investigation in the revised Research Implications section. (Lines 327–331).

Comments 4: Growth curves based on Hadlock centiles are among the least sensitive for diagnosing FGR; the most recommended ones remain the INTERGROWTH-21st and WHO growth standards.

Response 4: Thank you for highlighting this important point. While some studies suggest that newer standards such as INTERGROWTH-21st and WHO fetal growth charts may offer advantages in detecting fetal growth restriction (FGR), particularly in specific populations, the optimal choice of growth standard remains context-dependent. As recommended by FIGO guidelines (Melamed et al., 2021), institutions are encouraged to use fetal growth charts that have been validated in their local populations. In our setting, the Hadlock chart has undergone such validation and has demonstrated good predictive performance for adverse perinatal outcomes in a tertiary care context (Mendoza-Carrera et al., 2021). We have clarified this rationale in the revised Methods section. (Lines 109–112).

Comments 5: Was Doppler velocimetry performed in fetuses with an EFW between the 10th and 20th percentile?

Response 5: We confirm that Doppler velocimetry was performed in cases with suspected FGR per standard protocol. However, Doppler assessments were not performed for other cases. We have clarified this point in the Methods section (Lines 113–119).

Comments 6: How frequently and by what methods were these fetuses monitored?

Response 6: Thank you for this important question. Fetuses in the 10th–20th percentile group were monitored according to institutional protocols, with surveillance strategies individualized based on clinical judgment, maternal comorbidities, and evolving fetal findings. Follow-up assessments typically included serial ultrasounds for growth and amniotic fluid volume, with Doppler studies performed when clinically indicated as mentioned above. We have now clarified these monitoring practices in the revised Methods section (Lines 113–119).

Comments 7: Over 90% of the fetuses were delivered at term. Were any of these deliveries induced, and under what criteria?

Response 7: We have added this information to the Results section (Table 4).

Comments 8: In the subgroup of fetuses between the 10th and 20th centile, more than 37% were diagnosed as either FGR (68 cases; 21.1%) or SGA (53 cases; 16.4%). At what gestational age were these diagnoses made, and how were these fetuses managed—particularly regarding the timing of delivery?

Response 8: Thank you for your observation. FGR and SGA were diagnosed based on the criteria established by ISUOG. As shown in the updated Table 3, the timing of diagnosis varied among cases. Decisions regarding the timing of delivery were made by the attending obstetricians, following institutional protocols and current clinical guidelines. We have expanded and clarified this information in the revised Methods section (Lines 119–121).

Comments 9: The most immediate consideration is that if approximately 37% of fetuses with biometric measurements between the 10th and 20th percentile are later classified as FGR or SGA, the risk of adverse outcome increases significantly. Much of this risk likely depends on the methods used for fetal surveillance and the timing and mode of delivery.

Response 9: Thank you for highlighting this critical point. As noted in our findings, approximately 37.5% of fetuses with an EFW between the 10th and 20th percentiles were subsequently diagnosed with FGR or SGA, according to ISUOG criteria. This observation supports the idea that this “low-normal” group may harbor a subset of fetuses with early or evolving placental insufficiency and an increased risk of adverse outcomes.

We agree that these risks are not solely driven by fetal size but are also influenced by how fetal surveillance is conducted and how clinical decisions—particularly regarding delivery timing and mode—are made. This applies not only to the 10th–20th percentile group but also to fetuses below the 10th percentile, where effective monitoring and timely intervention are equally critical for optimizing outcomes.

As detailed in the revised Methods section, both groups in our study were followed using individualized surveillance strategies based on maternal comorbidities and evolving fetal parameters. Decisions regarding delivery were made by attending obstetricians in accordance with institutional protocols and current guidelines.

While our findings highlight the clinical relevance of this intermediate EFW group, we emphasize in the revised Discussion (Lines 319–331) (Lines 361–369) that further prospective studies are needed before any change in practice is recommended. Future research should aim to validate these observations in diverse populations and assess whether standardized monitoring protocols can improve detection and outcomes in this at-risk population.

Round 2

Reviewer 1 Report

Comments and Suggestions for Authors

I have no further comments

Comments on the Quality of English Language

I have no further comments

Author Response

.

Reviewer 2 Report

Comments and Suggestions for Authors

I feel comfortable with your response. I am still amazed that the 66.6% cesarean section rate in the EFW 20-90% group. By reviewing the table, I found out most of the C/S were elective without a clear indication (or not presented). This might be the result of local practice style, but it could be discussed if possible. Although the study did not reach the predetermined sample size, the post-hoc power analysis is good enough. 

Author Response

Comments 1: I feel comfortable with your response. I am still amazed that the 66.6% cesarean section rate in the EFW 20-90% group. By reviewing the table, I found out most of the C/S were elective without a clear indication (or not presented). This might be the result of local practice style, but it could be discussed if possible. Although the study did not reach the predetermined sample size, the post-hoc power analysis is good enough.

Response 1: Thank you for this observation. We agree that the high cesarean section rate, reflects local practice patterns. This point, along with the potential for intervention bias due to local management approaches, has now been explicitly addressed in the revised Limitations section of the manuscript. We also note that such institutional practices should be considered when interpreting the generalizability of our findings. Lines (360-370).

Reviewer 3 Report

Comments and Suggestions for Authors

Following modifications, particularly to the methodology and discussion, the study has been made more readable and clear. This makes it more interesting for a timely and open discussion on the management of SGA-FGR fetuses.
Congratulations to the authors.

Author Response

.